# microRNAs as Biomarkers of Breast Cancer

**DOI:** 10.3390/ijms26094395

**Published:** 2025-05-06

**Authors:** Wojciech Jelski, Sylwia Okrasinska, Barbara Mroczko

**Affiliations:** 1Department of Biochemical Diagnostics, Medical University, Waszyngtona 15 A, 15-269 Bialystok, Poland; barbara.mroczko@umb.edu.pl; 2Department of Biochemical Diagnostics, University Hospital, Waszyngtona 15 A, 15-269 Bialystok, Poland; sylwia.okrasinska@uskwb.pl; 3Department of Neurodegeneration Diagnostics, Medical University, Waszyngtona 15 A, 15-269 Bialystok, Poland

**Keywords:** microRNAs, breast cancer

## Abstract

Breast cancer (BC) is the most common type of cancer found in women. Detection of this cancer at an early stage is essential for effective treatment and a favorable prognosis. Potential early breast cancer biomarkers useful for diagnosing these tumors are microRNAs. These are small single-stranded RNA chains that can regulate the post-transcriptional expression of many different oncogenes. Cancer cells contain miRNAs that play a special role in the etiology of cancer development. The role of microRNAs in the initiation and development of breast cancer gives us great hope for the creation of molecular tools for early cancer detection. MicroRNAs are characterized by a high stability due to RNase, which protects them from degradation and enables their detection in various biological fluids. Researchers have described multiple serum microRNA signatures useful for detecting breast cancer. This review discusses the importance and potential usefulness of microRNAs in detecting breast cancer at an early stage, predicting the course of the disease, and assessing the effectiveness of treatment.

## 1. Introduction

Breast cancer (BC) is the most common malignancy affecting women, the leading cause of cancer death, and the leading cause of all-cause death in women aged 40–55 years. As a result, it remains a serious global health problem worldwide. Although significant progress has been made in its diagnosis and treatment, the heterogeneity of breast tumors and their varied responses to therapy necessitate a personalized approach to prognosis and treatment selection [1,2]. Breast cancer is a complex disease characterized by excessive cell growth and proliferation. Its etiology is multifactorial, involving both environmental and genetic factors, which results in different molecular subtypes with different clinical courses. With the use of mammography, more and more non-malignant lesions are detected, such as ductal carcinoma in situ, atypical ductal hyperplasia, lobular carcinoma in situ, and atypical lobular hyperplasia. These lesions are one of the most important risk factors for progression to invasive carcinogenesis [3,4]. Detecting cancer lesions at the earliest possible stage and assessing their prognosis are important for increasing the effectiveness of treatment for women with breast cancer. However, mammography is not the best early detection tool due to its limited sensitivity and specificity [5,6]. In recent decades, many new tumor markers have been described which take into account histological variability, tumor size, and the stage and status of axillary lymph nodes. These new potential biochemical markers include epidermal growth factor receptors, progesterone receptors, estradiol receptors, the erbB-2 proto-oncogene, and some proteolytic enzymes. However, none of these biochemical markers surpasses the status of the axillary nodes as a prognostic marker [7]. This is one of the reasons why microRNAs (miRNAs) have attracted attention in recent years as a new potential screening biomarker for breast cancer. Numerous studies have shown that microRNAs are aberrantly expressed in the tissue and blood of women with breast cancer. It has been found that miRNA dysregulation is quite different in normal breast tissue, precancerous tissue, and cancerous tissue, suggesting a role for microRNAs as potential effective biomarkers in differentiating between these stages. MicroRNA expression can be measured from various biological samples (tissue, blood), which increases its diagnostic and prognostic utility [8,9,10,11,12].

As the Food and Drug Administration (FDA) and European Medicines Agency (EMA) are adopting new treatment modalities, these institutions require accurate biomarkers to monitor disease progression and treatment efficacy. While clinical biomarkers are well-established and helpful in characterizing disease progression, there is a critical need for more robust and sensitive circulating biomarkers, such as nucleic acids and other small molecules. Used alone or in combination with clinical biomarkers, they will play a critical role in improving stratification for clinical trials and patient access to approved treatments, as well as in tracking responses to therapy, paving the way for the development of individualized therapeutic approaches. A biomarker is defined as an objectively measured parameter that serves as an indicator of normal biological processes, pathological processes, or biological responses to a therapeutic intervention (as defined by the FDA or EMA). Biomarkers can be divided into susceptibility/risk assessors; diagnostic, monitoring, prognostic, predictive, response, or safety biomarkers; and pharmacodynamic biomarkers.

Currently, mammography, ultrasound, magnetic resonance imaging, and tissue biopsy are the main diagnostic techniques available for detecting breast cancer. However, their expensive setup, need for skilled supervision and expert analysis, and invasiveness are their major limitations. Due to their high cost, these screening tests are beyond the reach of people belonging to low socio-economic groups, which leads to a serious health burden for society. The determination of microRNAs associated with breast cancer in a large population is cost-effective, non-invasive, and high-throughput. It will help to assess the risk of disease at an early stage, even in less-developed areas, and will also help to reduce the disease burden in society and the cost of treatment for common people.

MicroRNAs are non-coding small RNA molecules, usually composed of 15–25 nucleotides, that play a significant role in the regulation of various genes. They regulate genes by binding to complementary sequences on target messenger RNAs, resulting in their degradation or translational repression. They also play an important role in carcinogenesis and cancer metastasis. They are often factors involved in epigenetic modifications, and they participate in protein degradation and gene silencing, which are components of their physiological regulatory functions. Therefore, the deregulation of these non-coding RNAs is strongly associated with both procarcinogenic activity and, later, cancer progression [13]. Several different types of non-coding RNAs have been described, such as long non-coding RNAs (lncRNAs), piwi-interacting RNAs (piRNAs), and microRNAs (microRNAs). The regulatory functions of these microRNAs are diverse and vary depending on their subtype [14,15]. The expression of microRNAs is more significant in patients with positive hormone receptors and lower in the basal-like subtype. Before and after chemotherapy, microRNA expression is strongly associated with the expression of the BRCA1 and BRCA2 genes, as well as with the expression of the p53 gene. The same is not observed for HER2. Furthermore, the detection limits of HER2 are much lower than those for microRNAs.

In this systematic review, we have undertaken the task of comprehensively discussing the diagnostic utility of microRNAs in breast cancer. We have conducted an in-depth online search of articles from various databases such as Web of Science, Embase, and Pubmed. Combining these databases produced good results, achieving an overall recovery rate of over 96%. Ultimately, eighty published systematic reviews were included out of a total of 1930 relevant references identified by searching our databases (Figure 1). The search was conducted using the subject terms “breast cancer”, “microRNA”, and “diagnostic utility”. Two investigators (WJ and SO) independently extracted data from eligible articles that met the inclusion and exclusion criteria. Discrepancies were then reviewed and resolved through discussion. Studies were included in the meta-analysis if they met the inclusion criteria. We adopted inclusion criteria for the articles analyzed as being relevant by our full-text assessment, that is, articles concerning the association between microRNA expression levels and the detection of breast cancer. Studies were considered eligible if they met the following criteria: the diagnostic capacity of miRNA towards breast cancer was discussed; all women with cancer were diagnosed by the gold-standard test (histological examinations); and false positive, true positive, false negative, and true negative results were provided to construct a 2 × 2 contingency table. Articles were excluded based on the following criteria: being written in a language other than English; not being conducted on humans; being reviews, letters, and meeting records; breast cancer and microRNAs were not studied; studies focusing on gene polymorphisms; studies with insufficient data.

## 2. Function of microRNAs

A new mechanism mediated by microRNA molecules which influences the epigenetic regulation of gene expression was discovered at the beginning of the 21st century. Since then, microRNA research has intensified significantly, which clearly demonstrates their great biological importance and has led to the discovery of more and more non-coding microRNAs. According to the miRBase miRNA sequence database (release 22.0, March 2018), the human genome contains 2654 mature miRNA sequences [16]. Numerous miRNAs are highly conserved across mammalian species and have the ability to regulate important biological processes, including cell development, differentiation, aproliferation, stress responses, metabolism, and cell death differently, given the sequence. This is due to the fact that multiple microRNAs can target the same mRNA molecule and one miRNA molecule regulates multiple targets, which directly affects the number of translated proteins available, as shown by microarray methods [17]. MicroRNA biosynthesis begins with the transcription of dedicated information units by RNA polymerase II to the formation of target functional regulators through successive maturation steps. Each of these steps is controlled by many regulators in healthy cells. Some miRNAs are generated mainly through the canonical pathway, but others are generated within the canonical pathway. The regulation of gene expression by microRNA occurs through the binding of this target miRNA to the 3′ non-coding region (3′-UTR), which leads to the deadenylation of the microRNA or translational repression. MicroRNAs have been shown to be responsible for regulating over 60% of microRNA transcripts in various species. MicroRNA molecules present in various body fluids, such as blood, exhibit high stability. They have also been found in the lumen surrounding circulating extracellular vesicles (exosomes) [18,19]. Control of individual gene expression allows microRNAs to play a key function in physiological and pathophysiological processes, including responses to many environmental factors. Changes in the expression level and function of microRNAs are associated with many disease processes; metabolic disorders, including neurodegenerative disorders; viral infections; immune system diseases; cardiovascular diseases; and carcinogenesis.

## 3. The Role of microRNAs in Carcinogenesis

Our understanding of the etiology of breast cancer has improved over the past few years, but not all aspects have been clarified and it remains a challenge for researchers. Although the genetic basis of breast cancer does not completely explain its molecular anomalies, it is believed that the genomic causes responsible for the process of carcinogenesis are less often inherited and more often acquired. It is widely believed that epigenetic processes such as DNA methylation and histone modification and microRNAs are involved in every aspect of breast cancer’s pathophysiology [20,21,22,23]. The role of microRNAs in the pathogenesis of cancer, including breast cancer, is coming to be understood better. MiRNAs are still being widely studied for their role in specific biological processes, such as proliferation, cell differentiation, and apoptosis. This has shown that microRNAs play a key role in cancer etiogenesis, where their dysregulated expression is associated with tumor initiation, progression, and metastasis. In the case of breast cancer, specific miRNAs, such as miRNA-200c, play an important role in modulating the important processes of angiogenesis, cell proliferation, apoptosis, and adhesion. Disruption of the expression of these miRNAs results in a cascade of disorders in cells, including changes in the cell cycle and uncontrolled tumor growth [24,25]. Cancer-related miRNAs generally fall into two categories. The first group of oncogenic microRNAs (oncomiRs) includes highly expressed miRNAs. They are responsible for tumor progression and are important in maintaining the tumor phenotype. Tumor-suppressive miRNAs (miRsupps) belong to the second group and inhibit carcinogenesis by regulating cell growth, apoptosis, immune cell proliferation, and other factors that affect tumorigenesis and can often be downregulated in different tumors [26,27]. Some cancer-related miRNAs are also known as context-dependent miRNAs because they can exhibit specific effects dependent on the tumor tissue, i.e., a single miRNA can have oncogenic or tumor-suppressive effects in different tumors. An example of a dual-function microRNA is miR-29, which suppresses lung tumors and cellular lymphomas but is oncogenic in breast cancer [28,29]. By altering the expression of tumor suppressor genes and oncogenes, microRNAs can inhibit or exacerbate cancer progression depending on whether their expression is increased or decreased. This dual function highlights their potential as therapeutic targets as well as diagnostic tools. In this regard, specific microRNAs whose expression is reduced in atypical ductal hypoplasia but increased in invasive ductal carcinoma could be considered early warning signals for cancer progression. Wang et al. were the first to perform a large-scale bioinformatic analysis of human oncogenic and suppressive microRNAs examining their different functions, gene expression, evolutionary rate, molecular size, free energy, and transcription factors. They showed that oncogenic microRNAs, more frequently than suppressive microRNAs, cleave target mRNAs in tumor tissues. Moreover, oncomiR coding sequences were observed primarily in amplified chromosomal regions, in contrast to miRsupp sequences, which were more frequently present in deleted chromosomal regions [30,31]. However, although it has been confirmed that oncogenic microRNAs are upregulated and suppressive microRNAs are downregulated or disappear in cancer patients, the expression of some traditional oncomicroRNAs in patient blood decreased during the course of breast cancer. These results indicate that it is not possible to precisely define the oncogenic or suppressive properties of individual microRNAs [32]. MiRNA expression profiles in various human cancers showed a common reduction in the pool of microRNAs relative to the microRNAs present in normal tissues. Many cancer types had distinct miRNA expression profiles that could be used to distinguish the tumor type or tissue origin of poorly differentiated tumors. The finding of multiple isoforms of microRNAs associated with cancer confirms that these isomiRNAs play an essential role in miRNA-mRNA regulatory networks and that changes in the expression profiles of isomicroRNAs lead to cancer development.

It has been proven that genetic deletion/amplification, the methylation of microRNA loci, and changes that affect the regulation of primary microRNAs and transcription factors actively involved in the microRNA biosynthesis pathway commonly cause changes in microRNA expression and function in various cancers [22,33]. The regulatory action of microRNAs is mainly exerted by them binding to messenger RNAs within the 3′ non-coding region of their target genes, thus allowing them to influence gene expression post-transcriptionally. This interaction is crucial in regulating a range of biological processes essential for maintaining normal cellular functions [18].

The main enzyme involved in the expression of microRNAs is Dicer. Recent studies have shown that Dicer expression not only increases in advanced cancer tissue, but also that it controls metabolism in cancer tissues, thereby stimulating cell progression [34].

In different stages of mammary gland development, the expression level of microRNAs may be a critical regulator of tumor development. There is a documented association between microRNA levels and breast development, lactation, or carcinogenesis. MicroRNAs such as microRNA-126, microRNA-150, and microRNA-145 have been shown to play a role in lipid metabolism during lactation [35].

MicroRNAs play a key role in the spread of metastatic cancer cells. This includes the processes of invasion and migration, changes in the microenvironment, and the development of a poor prognostic phenotype. These microRNAs are termed metastamicroRNAs and have been both upregulated and downregulated in various cancers, including breast cancer. It has been shown that microRNA-126 and microRNA-126* can modify the tumor micro-environment, thereby inhibiting breast cancer metastasis through both the microRNA-dependent suppression of mesenchymal stem cells and the aggregation of inflammatory monocytes in the tumor stromal environment [36,37]. Additionally, miR-494 was shown to have the ability to inhibit the CXCR4 receptor and shown to be downregulated in breast cancer cells. This microRNA also contributes to cancer inhibition via the CXCR4-dependent Wnt/-catenin signaling pathway [38].

A significant cross-correlation was also demonstrated between the expression of the progesterone receptor and the expression level of some microRNAs, such as microRNA-181a, microRNA-23a, and microRNA-26b, suggesting their involvement in the regulation of this receptor over the course of breast cancer. Additionally, the changes in the expression of these three microRNAs in tumor tissues compared to normal tissues showed an opposite trend during the analysis of tumor immunohistochemical status, confirming their role in the development of breast cancer [7].

## 4. microRNAs as Biomarkers of Breast Cancer

Breast cancer is one of the most serious health problems in women worldwide, with a high mortality rate each year. The detection of precancerous or early asymptomatic cancers is important, due to the effectiveness of treatment at this stage. Many methods, such as breast self-examination, mammography, and magnetic resonance imaging, aim to detect breast cancer as early as possible [39,40,41]. However, it should be remembered that the sensitivity of all currently used methods is limited and that, especially in younger women, mammography results are often difficult to interpret due to high-density breast tissue [42,43,44,45]. Effective and early blood biomarkers, as a complement to imaging methods for breast cancer detection, have been widely explored recently. A reliable diagnostic test may not only increase the diagnostic power of breast cancer screening and support the need for further diagnostic imaging procedures, but it may also increase patient compliance and the acceptance of preventive medical check-ups. Potential candidates for molecular markers that would fulfil these hopes are microRNA expression signatures. Liquid biopsy in women with breast cancer is a minimally invasive procedure involving microRNA profiling, which can provide data on the expression of microRNAs in various types of cancer and at various stages of the disease. This allows for observation of the course of the disease in real time and from a longitudinal perspective and allows for the monitoring of the response to treatment [46,47,48]. Measuring miRNA expression in various biological materials, such as blood, plasma, and tissue, further increases the possibility of its use for both diagnostic and prognostic purposes (Figure 2). MicroRNAs can be studied in very small amounts of sample and in altered materials. An additional advantage is their high stability in tissues and fluids. This allows for less invasive sampling methods, which makes microRNAs particularly attractive for routine clinical use. It is clear that tumor-derived microRNAs, as freely circulating markers, can be measured in the same way as miRNAs and from various types of blood cells. The progress in molecular methodologies observed in recent years has allowed for the use of microRNA microarrays to identify panels of microRNAs that distinguish between different types of cancer cells with greater sensitivity [49,50,51,52,53,54,55].

Research conducted in recent years has expanded our knowledge about specific microRNAs useful in the diagnosis of breast cancer (Table 1). Verma et al., examining samples of different types and stages of breast cancer using two separate array platforms, identified about 440 microRNAs associated with this cancer. Their hierarchical clustering and molecular features strongly suggested distinct or common microRNA signatures. Of these, 107 microRNAs could be classified as potential biomarkers for detecting different types and stages of breast cancer. These observations were further confirmed by Northern blot analyses and tissue microarrays of the same samples [56]. In turn, studies by Sochor et al. and Li et al. [57,58] have shown significant differences in the expression profiles of microRNAs between different stages of breast cancer. In particular, their studies have shown that microRNA-19a, microRNA-155, microRNA-24, and microRNA-181b, which have tumor suppressor gene inhibition properties, are significantly elevated in serum samples from cases of ductal carcinoma in situ and invasive ductal carcinoma compared to healthy tissue and atypical ductal hyperplasia. Of particular interest is microRNA-155, which can be inhibited by the proven breast cancer suppressor proteins BRCA1 and BRCA2. Additionally, microRNA-155 promotes the proliferation and migration of breast cancer cells by downregulating the expression of the suppressor of cytokine signaling 1 and upregulating the expression of matrix metallopeptidase 16. On the other hand, microRNA-206, microRNA-571, microRNA-193a-3p, microRNA-519a, and microRNA-526b, which typically inhibit oncogenes, were significantly downregulated in plasma samples from women with ductal carcinoma in situ, invasive ductal carcinoma, or atypical ductal hyperplasia compared to healthy tissue. These results indicate the potential of microRNA expression in blood being a useful biomarker of breast cancer. These microRNAs may aid in the diagnosis process and, as a result, show promise in distinguishing between different stages of breast cancer, such as ductal carcinoma in situ, invasive ductal carcinoma, or atypical ductal hyperplasia, especially when used in conjunction with tissue biopsies for confirmation [57,58,59]. MicroRNA-21, whose expression increases in women with breast cancer and is known to affect tumor suppressor genes, contributing to tumorigenesis, may be of great use. MicroRNA-21 has been significantly expressed in triple-negative breast cancer (TNBC) tissue and has been associated with increased TNBC cell invasion and proliferation and reduced PTEN expression [60]. In turn, the change in the expression of microRNA200c and microRNA-205 accompanies the epithelial-to-mesenchymal transition, a critical process in cancer metastasis [61]. Five different microRNAs in the serum of breast cancer patients (microRNA-21, microRNA-125b, microRNA-145, microRNA-155, and microRNA-365) were studied by Han et al. They examined the levels of microRNAs in approximately 100 women with breast cancer and 21 healthy controls. In addition, microRNA expression was measured in 20 patients after surgical resection of their tumor. It was found that the expression of microRNA-155 in stages I and II was significantly higher compared to in stage III. Furthermore, the levels of microRNA-21 and microRNA-155 were significantly decreased after surgical resection. A ROC curve analysis evaluating the sensitivity and specificity of these microRNAs as diagnostic biomarkers showed that the combination of serum microRNA-21, microRNA-155, and microRNA-365 could be used as a sensitive and specific biomarker to distinguish breast cancer patients from healthy individuals [62]. The ROC curve analysis confirmed the usefulness of microRNA in the studied samples for predicting disease recurrence in breast cancer. The combination of several microRNAs led to higher accuracy compared to each microRNA alone. Furthermore, the inclusion of clinicopathological prognostic factors significantly improved the ability of microRNAs to stage the disease [63,64].

The micoRNA-200 family, which includes microRNA-200a, 201b, 201c, microRNA-141, and microRNA-429, has opposing roles in regulating epithelial–mesenchymal transition (EMT) and metastasis. They may negatively regulate E-cadherin transcriptional repressors ZEB1/2, preventing epithelial–mesenchymal transmission, but they are also associated with global changes in gene expression that promote the metastatic colonization of breast cancer. These opposing data on the clinical significance of miR-200 family members in breast cancer have been observed. As a result of their comprehensive profiling approach, Madhavan et al. showed that microRNA-200b and microRNA-200c are among the six microRNAs highly expressed in women with early breast cancer who developed metastases [65,66,67,68]. These results were confirmed by Papadaki et al., who demonstrated an association between the plasma microR-200 family and metastatic progression in breast cancer. Moreover, a strong expression of microRNA-200c corresponds to late relapse and is an independent prognostic factor for worse disease-free survival [69]. Fischer et al. showed that serum levels of circulating microRNA-200s associated with EMT were increased before therapy (microRNA-200a, microRNA-200b, and microRNA-141) and during disease progression (microRNA-200a, microRNA-200b, microRNA-200c, microRNA-141, and microRNA-429) compared to their levels after one cycle of systemic therapy [70]. These results are consistent with previous reports demonstrating increased expression levels of microRNA-200c and miR-141 during tumor progression and metastasis [71]. Furthermore, increased serum levels of microRNA-200b and microRNA-200c accurately distinguished patients with early stage breast cancer from those with metastatic breast cancer. Interestingly, the results showed increased serum levels of microRNA-200b in premenopausal breast cancer patients compared to postmenopausal breast cancer patients [69]. The present results are consistent with the current literature, indicating that all circulating microRNA-200s are upregulated during metastatic disease. MicroRNA-503, whose expression is reduced in breast cancer cells and whose overexpression reduces cell proliferation by targeting cell cycle regulation, has also been shown to be overexpressed in the breast cancer tissue and blood plasma of breast cancer patients compared to healthy tissue. This overexpression of microRNA-503 in breast cancer cells inhibits the expression of the protein associated with epithelial–mesenchymal transition, SMAD2, and the epithelial marker protein E-cadherin. It has been shown that microRNA-503 regulates the oncogene ZNF217 and that high expression of this microRNA leads to improved survival over the course of breast cancer. MicroRNA-503 acts as a tumor suppressor via the DDHD2 gene in breast cancer cells [72]. MicroRNA-1307 is also overexpressed in women with breast cancer. MicroRNA-1307 was found to be differentially expressed, with a fold-change of 0.36, between breast cancer and adjacent normal parenchymal tissue. The overexpression of this microRNA is associated with BRCA1 in breast cancer compared to normal tissue counterparts [73,74,75,76].

MicroRNAs can be released from exosomes. Exosomes have been detected in all body fluids, including blood. To adequately describe the composition of exosomes obtained from body fluids or tissues, they must first be isolated while preserving their structure and then examined for their size, morphology, biochemical composition, and cellular origin. Currently, the clinical translation of this method is hampered by the lack of gold-standard techniques for the rapid isolation, purification, and quantification of exosomes. Several methods are available for isolating exosomes, such as ultracentrifugation and size-based techniques, precipitation methods, immunoaffinity techniques, and microfluidics-based methods. The obtained exosome fraction yields vary in purity and size depending on the method used. The relevance of the approach depends on the sample’s source and the intended use of the exosomes. Moreover, additional challenges include ensuring the reproducibility and consistency of the resulting exosome isolates and adequate quality control and standardization across research groups.

Based on the literature data, Itani et al. investigated the utility of the expression of microRNA-21, microRNA-155, microRNA-23a, microRNA-130a, microRNA-145, microRNA-451, microRNA-425-5p, microRNA-139-5p, microRNA-195, microRNA-125b, microRNA100, and microRNA-182 in different ethnic groups for the diagnosis of breast cancer. The level of these twelve selected microRNAs was evaluated in the plasma of women with early-stage breast cancer of the most common histotype and receptor profile in Lebanon. Then, a comparison was made with healthy controls to check the correlation between the clinical data and their diagnostic ability. They showed that microRNA-21, microRNA-155, and microRNA-23a were upregulated in plasma from Lebanese women with early-stage BC compared to controls, which was similar to the results seen in other ethnic groups. While the upregulation of microRNA-21 and microRNA-155 expression in serum from Chinese ethnic groups was found previously, patients in the study by Itani et al. confirmed these findings and that the overexpression of these microRNAs plays an oncogenic role in the development of breast cancer. In turn, the expression of microRNA-130a, microRNA-145, and microRNA-451 in plasma from Lebanese women with early-stage BC behaves differently from that in other ethnic groups. MicroRNA-145 and microRNA-130a were increased in Lebanese patients and decreased in tissue and plasma samples from other ethnic groups. In contrast, microRNA-451 is downregulated in the plasma of Lebanese women but upregulated in the plasma of Chinese ethnic groups [77].

Jing et al. conducted a high-impact study on 90 patients and 65 healthy individuals, investigating eight microRNAs associated with breast cancer (including microRNA-10b-5p, microRNA-133a-3p, microRNA-195-5p, microRNA-195-3p, and microRNA-155-3p). They found that the five microRNAs listed above offer promise for the early, noninvasive, and accurate diagnosis of breast cancer. This was the first study of these microRNAs to demonstrate their strong ability to distinguish breast cancer from benign breast disease (e.g., microRNA-133a-3p has an area under the ROC curve of 0.84) [78].

MicroRNAs have the potential to predict the response of breast cancer to systemic treatments. This includes the expression of microRNA-342-3p and microRNA-187-3p, which was linked to the success of systemic treatment. High expression levels of miRNA-342-3p and miRNA-187-3p support progression-free and overall survival [79].

The sensitivity and specificity of a panel of microRNA-4443, microRNA-572, and microRNA-150-5p, and each of these microRNAs separately, were investigated in terms of their potential for use as diagnostic biomarkers of breast cancer. The data obtained from the analysis showed that microRNA-4443 and microRNA-572 can be considered excellent biomarkers, while microRNA-150-5p can be considered a good biomarker in the diagnosis of breast cancer. In addition, a logistic regression analysis was performed to evaluate the diagnostic potential of a panel of these three microRNAs. The panel of these microRNAs achieved a specificity of 97.22% and a sensitivity of 76.67%, with an AUC of 0.9366 (*p*-value < 0.0001). These results indicate the strong diagnostic potential of this miRNA panel in diagnosing patients with breast cancer [80]. It should be noted that the sensitivity of CA15-3 (a recognized biomarker of breast cancer) is 63.3%, while its specificity is 60.7% and its AUC is 0.64 [81].

There are many methods available for analyzing microRNAs, but the most commonly used methods are microarrays and next-generation sequencing. MicroRNAs are isolated from a sample and further analyzed based on specific medical indications [82,83].

## 5. Future Perspectives and Conclusions

Breast cancer is a very large medical, social, and economic problem. Therefore, finding useful diagnostic tools such as early biomarkers to achieve positive treatment outcomes for patients is a big challenge. It is obvious that breast cancer research requires prospective action through large-scale randomized prospective trials. Numerous studies have clearly demonstrated the potential effectiveness of microRNAs in the diagnostic and therapeutic processes of breast cancer. microRNAs regulate numerous mechanisms involved in the process of breast cancer carcinogenesis, such as cell proliferation, invasion, and apoptosis. The diagnostic relevancy of microRNAs is one of the most attractive and modern research areas related to these molecules in oncology [84]. Recently, there have been great advances in the microRNA profiling of liquid biopsy samples. Some microRNAs have been identified as candidate biomarkers, allowing for the detection of early-stage breast cancer and for it to be distinguished from benign lesions in the mammary gland using a small amount of blood. Paying close attention to the research on the microRNAs involved in modifying the biological pathways that cause the process of breast cancer carcinogenesis is an appropriate approach to discovering a minimally invasive test for the early detection of biomarkers in women with breast cancer in its asymptomatic stages or with a negative CA15-3 result. The creation of a CA15-3 assay panel in combination with microRNA or other proteomic markers for the diagnosis of breast cancer may increase the sensitivity and effectiveness of these markers. MicroRNA extracted from various biological fluids (blood, serum, plasma, etc.) is characterized by a high stability, which is why it can be used as a biomarker for more accurate and less invasive diagnostics, improving the effectiveness of therapy. Further development of our knowledge about microRNA could help promote new diagnostic and therapeutic strategies. Some researchers claim that unfortunately the sensitivity and specificity values of a single marker are too low to be satisfactory for clinical needs when it comes to diagnosing diseases. On the other hand, some studies have shown that the combined analysis of several microRNA targets results in higher accuracy and sensitivity and a larger area under the ROC curve, making it more diagnostically useful [85].

Network analysis allows for more information to be gained by incorporating data on the expression of the protein-coding microRNAs associated with the miRNAs involved in cancers into its analysis. Using a model of intervention efficiency based on a combination of microRNAs, complementary information orthogonal to that obtained from pathological characteristics can be obtained.

Unfortunately, different microRNA studies have various limitations. In some, the validation of microRNA biomarkers is insufficient, probably due to the difficulty in distinguishing between closely related microRNAs, the lack of standardized normalization procedures, and differences between individual methods. Difficulties in detecting microRNAs may also result from their small size, which requires specialized and dedicated analytical tools. Furthermore, during the storage of blood samples, microRNAs can be released from erythrocytes or leukocytes, which can falsify the results. In addition, leukocytes and hemolysis negatively affect the quality and quantity of the microRNA obtained. The level of microRNA in the blood depends on numerous factors, including gender, age, and lifestyle. Knowledge of the influence of individual factors on microRNAs allows us to learn the characteristics of the diagnostic values of microRNAs measured in a specific environment [86]. An important fact that influences the usefulness of microRNAs as a diagnostic biomarker is the correlation between the detection of certain microRNAs during the course of different types of cancer. Increased levels of microRNAs in blood have been found not only in women with breast cancer, but also in patients with esophageal, pancreatic, liver, colon, and lung cancer. However, results are sometimes inconsistent even in uniform studies of the same disease, which is another problem. Importantly, the reduced expression of microRNAs in cancer cells can be caused by genomic perturbations or the tumor adversely affecting the expression of microRNAs in other cells or reducing the stability of circulating microRNAs. This suggests that a reduced expression of microRNAs in serum may be explained as a nonspecific response to the presence of cancer [87]. Furthermore, microRNAs assayed by liquid biopsy may include other body fluids containing cancer-specific microRNAs.

Attention should be paid to the limitations that exist for a meta-analysis such as ours. First, considerable heterogeneity was observed in the meta-analysis, which may be due to the use of different cut-off values. Second, after applying our quality assessment to the diagnostic accuracy studies, it was found that all studies included in this meta-analysis were retrospective case–control studies. Furthermore, the results of the index tests were interpreted with a knowledge of the results of the reference standard, and the thresholds used were not specified in advance. Third, studies with positive results are more likely to be published, which may inflate the overall diagnostic accuracy of microRNAs. Fourth, we included only studies written in English, which may have influenced our conclusions.

Despite the great hope of microRNA being a useful biomarker of breast cancer, no microRNA has yet been used in practice for the diagnosis of patients with breast cancer. Studies on the usefulness of microRNA in the early detection of breast cancer are still being conducted. There is a need to develop new assumptions that will improve treatment effects and patient survival. The bulk of our attention should be focused on solving the current methodological and analytical difficulties and using automated and standardized operating procedures. It is also necessary to improve inter- and intra-method reproducibility. The improvement of breast cancer stratification methods using microRNA diagnostic kits should be seen as an improvement of the quality of current diagnostic techniques and lead to the development of personalized breast cancer treatment. MicroRNA profiling using a bead array may be of great utility for studying miRNA expressions in large-scale diagnostic studies due to its high throughput and cost-effectiveness. Throughout the diagnostic and therapeutic processes of breast cancer, interdisciplinary cooperation is particularly important in both diagnostic and therapeutic areas (oncology, gynecology, surgery, and radiotherapy). Further research is needed to explore the microRNAs associated with breast cancer [88,89]. These microRNAs could be used in combination with traditional biological diagnostic indicators for the clinical adjuvant screening of breast cancer at an early stage. As more breast cancer-related microRNAs are discovered, the potential for microRNAs-based breast cancer diagnosis and treatment will grow. For example, new gold nanoparticle carriers could provide microRNA delivery to cancer cells. This is a big challenge, but it is gradually being overcome thanks to advanced technologies, such as nanotechnology. Thus, microRNAs may soon be useful as new biomarkers and therapeutic targets for women with breast cancer [90,91].

The use of microRNA-34 modified by the conjugation of FM-miR-34a with folic acid (FM-FolamiR-34a) inhibits tumor growth, leading to a complete cure. These results demonstrate that we have the ability to revitalize microRNA-34a as an anticancer agent, providing a strong rationale for clinical trials [92]. Folate–miR-34a has potent inhibitory effects on FOLR1-expressing breast, ovarian, and cervical cancer cells, which are suggestive of its potential therapeutic application in FOLR1-expressing tumors. The results presented in a study by Li et al. highlight the challenges in the specific delivery of folate–miR-34a to cancer cells due to the lack of target receptor expression and shed light on the future development of ligand-coupled miR-34a as a potential therapy for advanced and aggressive cancer [93].

MicroRNAs are used in therapies and for testing responses to therapy. The two main therapies that are based on microRNAs are miRNA mimetics (i.e., miRNA replacement and restoration) and antagomiRs (i.e., miRNA reduction and suppression). However, researchers face limitations and challenges with the delivery of microRNA into cancer. There are various methods, such as systemic delivery systems (e.g., viral delivery, nonviral delivery (lipid-based microNA delivery; polymer-based microRNA delivery; atelocollagen-mediated systemic delivery), modified microRNAs (ASOs; LNA), and small transport domains (Ap-tamers)), being used to restore or suppress deregulated microRNAs.

It is common knowledge that chemotherapy and hormone therapy work through different mechanisms. Therefore, the systemic treatment of breast cancer patients should be determined according to the molecular subtype of their tumor. However, in the era of personalized therapy, more than one-third of patients with breast cancer do not respond to systemic therapy. Therefore, ongoing efforts should be made to identify biomarkers that will help identify patients who will benefit from systemic therapy. The detection of a single microRNA or a group of microRNAs can predict resistance to multiple therapeutic strategies.

The demonstrated potential of microRNAs as biomarkers for breast cancer opens the path for future clinical trials. If this could be further developed into a lab-on-a-chip system, it could prove to be a boon for women in low-income countries who are reluctant to undergo physical examinations. A better understanding of microRNA targets through in silico analysis, extensive validation, and massive clinical trials opens up the possibility of more refined, cost-effective, and non-invasive methods for breast cancer diagnosis able to reach the whole world’s population. The next steps include the rigorous validation of these microRNAs in larger groups of women, followed by the design and implementation of clinical trials to test their efficacy in clinical practice.

## Figures and Tables

**Figure 1 ijms-26-04395-f001:**
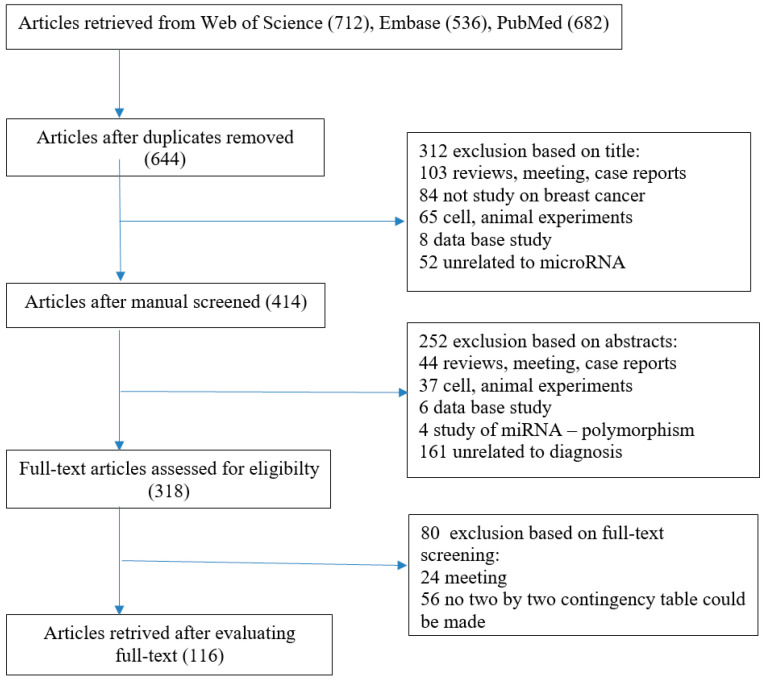
Flow diagram of the selection of studies for the present meta-analysis.

**Figure 2 ijms-26-04395-f002:**
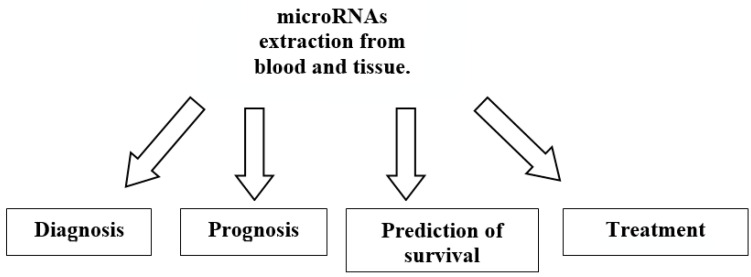
Role of microRNAs as biomarkers of breast cancer.

**Table 1 ijms-26-04395-t001:** Selected microRNAs differentially expressed in breast cancer.

Symbol	Expression	Materials	Function	Biomarker
microRNA-23a	Up	Plasma	Modify tumor micro-environment, metastasis	Prediction of survival
microRNA-200a	Down	Biopsy, Blood, Urine	Proliferation, invasion, metastasis	Diagnosis, prognosis
microRNA-200b	Down	Biopsy, Blood, Urine	Proliferation, invasion, metastasis	Diagnosis, prognosis
microRNA-200c	Down	Biopsy, Blood, Urine	Proliferation, invasion metastasis	Diagnosis, prognosis
microRNA-21	Up	Plasma	Proliferation, invasion, metastasis	Early detection, monitoring recurrences
microRNA-141	Down	Biopsy, Blood, Urine	Proliferation, invasion, metastasis	Diagnosis, prognosis
microRNA-429	Down	Biopsy, Blood, Urine	Proliferation, invasion, metastasis	Diagnosis, prognosis
microRNA-181b	Up	Blood	Proliferation, migration, invasion	Prediction of survival
microRNA-182	Down	Blood	Apoptosis	Diagnosis, prognosis
microRNA-125b	Up	Plasma, Serum	Proliferation, migration, invasion	Diagnosis, prognosis
microRNA-130a	Down	Plasma, Serum	Migration, invasion	Prediction of treatment response
microRNA-193a-3p	Down	Plasma	Proliferation, migration, invasion	Prediction of treatment response
microRNA-451	Up	Biopsy, Plasma	Proliferation, migration	Diagnosis, prognosis, prediction of treatment response
microRNA-331	Up	Blood	Proliferation, migration, invasion, metastasis	Diagnosis, prognosis, prediction of survival
microRNA-100	Up	Plasma	Proliferation,	Diagnosis, prognosis
microRNA-145	Down	Plasma	Cancer cell motility inhibition	Diagnosis, prognosis
microRNA-195	Up	Serum	Proliferation, migration	Diagnosis, prognosis
microRNA-155	Up	Plasma, Serum	Proliferation, metastasis	Prediction of survival, monitoring recurrences
microRNA-342-3p	Up	Plasma, serum	Apoptosis	Response to treatment
microRNA-187-3p	Up	Plasma, serum	Apoptosis	Response to treatment
microRNA-365	Down	Serum	Migration, apoptosis	Diagnosis, prognosis
microRNA-425-5p	Up	Plasma, Serum	Proliferation, migration, invasion	Prediction of survival

## Data Availability

No new data were created or analyzed in this study.

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
