# Peer review of "microRNAs as Biomarkers of Breast Cancer"

_ijms, 2025, doi:10.3390/ijms26094395_

Round 1
Reviewer 1 Report
Comments and Suggestions for Authors
The article titled “MicroRNAs as biomarkers of breast cancer” provides an overview of microRNAs (miRNAs) as biomarkers for early breast cancer detection, prognosis, and treatment monitoring. The authors summarize current research on miRNA biogenesis, dysregulation in carcinogenesis, and clinical applications while highlighting methodological challenges and ethnic variability in miRNA expression profiles. The review effectively summarizes miRNA biology and their oncogenic/tumor-suppressive roles, supported by robust references to mechanistic studies (e.g., miR-21 in triple-negative breast cancer, miR-155’s dual regulation). However, it inadequately addresses contradictory evidence, such as miR-200’s conflicting roles in metastasis, which undermines its diagnostic reliability. While the authors note ethnic disparities in miRNA expression (e.g., miR-145 differences in Lebanese vs. Chinese cohorts), they fail to explore implications for global biomarker applicability or population-specific screening protocols. The discussion of liquid biopsy potential is compelling but lacks concrete examples of validated clinical assays, relying instead on provisional panels like miR-21/miR-155/miR-365.
Methodological limitations are superficially treated: preanalytical variables (e.g., hemolysis), normalization strategies, and inter-platform variability (microarrays vs. sequencing) require deeper analysis. The review also overlooks cost-effectiveness comparisons with mammography or MRI, which are critical for real-world adoption. Although miRNA stability in biofluids is emphasized, there is no mention of extracellular vesicle (exosome) isolation techniques that could enhance specificity. The section on future perspectives correctly identifies the need for standardized protocols but offers no actionable frameworks, such as consensus guidelines from organizations like the FDA or ESMO. Finally, the therapeutic potential of miRNA mimics or inhibitors is underexplored, despite promising preclinical data on miR-503 and miR-1307. The reviewer has the following comments that need to be addressed by the authors:
- To provide a more comprehensive analysis, it would be helpful to address the conflicting findings regarding the dual roles of the miR-200 family. Discussing these discrepancies and their potential impact on clinical interpretation will offer a more nuanced perspective and strengthen the overall argument by acknowledging the complexities in the current literature.
- To clarify the clinical utility of miRNA-based approaches, it would be valuable to provide examples of miRNA panels currently being evaluated in active clinical trials. Additionally, comparing their sensitivity and specificity to established biomarkers such as CA15-3 would help highlight their potential advantages and limitations, thereby offering a more comprehensive view of their clinical relevance.
- To enhance the discussion on exosome-derived miRNAs, it would be beneficial to elaborate on the various exosome isolation techniques and their impact on improving biomarker specificity. A detailed exploration of these methods could provide valuable insight into how optimizing exosome isolation can enhance the reliability and precision of miRNA-based biomarkers in clinical applications.
- Expanding on the various approaches to miRNA-based therapeutics, such as antagomirs and miRNA mimics, would greatly enhance the review. Discussing their development in preclinical and clinical settings, as well as their potential therapeutic applications and ongoing trials, would provide a more comprehensive understanding of the progress and future potential of miRNA-based therapies.
- To strengthen the discussion on miRNA-based therapeutics, it would be beneficial to cite the given articles. They highlight the promising potential of miR-34a, highlighting its outstanding stability, activity, and anti-tumor efficacy. Including these references would complement the insights provided in this review on miRNAs as biomarkers for breast cancer and further emphasize the relevance of miRNA stability and therapeutic efficacy.
https://www.nature.com/articles/s41388-023-02801-8
https://www.mdpi.com/1422-0067/25/4/2123
- It would be valuable to evaluate the economic feasibility of miRNA testing in comparison to mammography, particularly in low-resource settings. Assessing the cost-effectiveness and accessibility of miRNA-based diagnostic approaches relative to traditional methods could provide important insights into their potential for widespread adoption in underserved populations. This addition would further strengthen the practical relevance of miRNA testing as a diagnostic tool.
- A discussion on the regulatory pathways for miRNA diagnostics would greatly strengthen the review. Proposing clear steps for FDA/EMA approval, including essential validation benchmarks like analytical and clinical performance criteria, would offer valuable guidance on how to bring miRNA-based diagnostics to market. This would further emphasize the practical and translational potential of miRNA technologies in clinical applications.
Author Response
Reviewer #1
1. To provide a more comprehensive analysis, it would be helpful to address the conflicting findings regarding the dual roles of the miR-200 family. Discussing these discrepancies and their potential impact on clinical interpretation will offer a more nuanced perspective and strengthen the overall argument by acknowledging the complexities in the current literature.
Response: Fischer et al. showed that serum levels of circulating microRNA-200s associated with EMT were increased before therapy (microRNA-200a, microRNA-200b, microRNA-141) and during disease progression (microRNA-200a, microRNA-200b, microRNA-200c, microRNA-141, and microRNA-429) compared to levels after 1 cycle of systemic therapy [71]. These results are consistent with previous reports demonstrating increased expression levels of microRNA-200c and miR-141 during tumor progression and metastasis [72]. Furthermore, increased serum levels of microRNA-200b and microRNA-200c accurately distinguished patients with early stage breast cancer from metastatic breast cancer. Interestingly, the results showed increased serum levels of microRNA-200b in premenopausal breast cancer patients compared to postmenopausal breast cancer patients [70]. The present results are consistent with the current literature indicating that all circulating microRNA-200s are upregulated during metastatic disease.
2. To clarify the clinical utility of miRNA-based approaches, it would be valuable to provide examples of miRNA panels currently being evaluated in active clinical trials. Additionally, comparing their sensitivity and specificity to established biomarkers such as CA15-3 would help highlight their potential advantages and limitations, thereby offering a more comprehensive view of their clinical relevance.
Response: The sensitivity and specificity of the panel and each of microRNA-4443, microRNA-572, and microRNA-150-5p as diagnostic biomarkers of breast cancer were investigated. The data obtained from the analysis showed that microRNA-4443 and microRNA-572 can be considered as excellent biomarkers, and microRNA-150-5p can be considered as a good biomarker in the diagnosis of breast cancer. In addition, logistic regression analysis was performed to evaluate the diagnostic potential of the panel of these three microRNAs. The panel of these microRNAs achieved a specificity of 97.22% and a sensitivity of 76.67%, with an AUC of 0.9366 (p-value < 0.0001). These results indicate a strong diagnostic potential of this miRNA panel in diagnosing patients with breast cancer [81]. It should be noted that for CA15-3 (a recognized biomarker of breast cancer) the sensitivity is in the order of 63.3%, specificity 60.7%, AUC 0.64 [82].
3. To enhance the discussion on exosome-derived miRNAs, it would be beneficial to elaborate on the various exosome isolation techniques and their impact on improving biomarker specificity. A detailed exploration of these methods could provide valuable insight into how optimizing exosome isolation can enhance the reliability and precision of miRNA-based biomarkers in clinical applications.
Response: MicroRNAs can be released from exosomes. Exosomes have been detected in all body fluids, including blood. To adequately describe the composition of exosomes from body fluids or tissues, they must first be isolated while preserving their structure and then examined for size, morphology, biochemical composition, and cellular origin. Currently, clinical translation is hampered by the lack of gold standard techniques for rapid isolation, purification and quantification of exosomes. Several methods are available for exosome isolation, such as ultracentrifugation and size-based techniques, precipitation methods, immunoaffinity techniques and microfluidics-based methods. The obtained exosome fraction yields varying purity and size depending on the method. The relevance of the approach depends on the sample source and the intended use of the exosomes. Moreover, additional challenges include the reproducibility and consistency of the resulting exosome isolates and adequate quality control and standardization across research groups.
4. Expanding on the various approaches to miRNA-based therapeutics, such as antagomirs and miRNA mimics, would greatly enhance the review. Discussing their development in preclinical and clinical settings, as well as their potential therapeutic applications and ongoing trials, would provide a more comprehensive understanding of the progress and future potential of miRNA-based therapies.
Response: MicroRNAs are used in therapies and testing for response to therapy. The two main therapies based on microRNAs are miRNA mimetics (i.e., miRNA replacement and restoration) and antagomiRs (i.e., miRNA reduction and suppression). However, researchers are faced with limitations and challenges for microRNA delivery into cancer. There are various methods such as systemic delivery system (e.g. viral delivery, nonviral delivery (lipid-based microNA delivery; polymer-based microRNA delivery; atelocollagen-mediated systemic delivery), modified microRNA (ASOs; LNA), and small transport domains (Ap-tamers)) to restore or suppress deregulated microRNAs
5. To strengthen the discussion on miRNA-based therapeutics, it would be beneficial to cite the given articles. They highlight the promising potential of miR-34a, highlighting its outstanding stability, activity, and anti-tumor efficacy. Including these references would complement the insights provided in this review on miRNAs as biomarkers for breast cancer and further emphasize the relevance of miRNA stability and therapeutic efficacy.
Response: The use of modified microRNA-34 by conjugation of FM-miR-34a with folic acid (FM-FolamiR-34a) inhibits tumor growth, leading to complete cure. The results demonstrate the ability to revitalize microRNA-34a as an anticancer agent, providing a strong rationale for clinical trials [93]. Folate–miR-34a has potent inhibitory effects on FOLR1-expressing breast, ovarian, and cervical cancer cells, suggesting its potential therapeutic application in FOLR1-expressing tumors. The results presented in the study by Li et al. highlight the challenges in specific delivery of folate–miR-34a to cancer cells due to the lack of target receptor expression and shed light on the future development of ligand-coupled miR-34a as a potential therapy for advanced and aggressive cancer [94].
6. It would be valuable to evaluate the economic feasibility of miRNA testing in comparison to mammography, particularly in low-resource settings. Assessing the cost-effectiveness and accessibility of miRNA-based diagnostic approaches relative to traditional methods could provide important insights into their potential for widespread adoption in underserved populations. This addition would further strengthen the practical relevance of miRNA testing as a diagnostic tool.
Response: Currently, mammography, ultrasound, magnetic resonance imaging and tissue biopsy are the main diagnostic techniques available for detecting breast cancer. However, expensive setup, requirement of skilled supervision, expert analysis, invasive procedure are their major limitations. Due to high cost, these screening tests are beyond the reach of people belonging to low socio-economic groups, which poses a serious health burden to the society. Determination of microRNAs associated with breast cancer in a large population is cost-effective, non-invasive and high-throughput. It helps to assess the risk of disease at an early stage even in backward areas and also helps to reduce the disease burden in society and cost of treatment for a common man.
7. A discussion on the regulatory pathways for miRNA diagnostics would greatly strengthen the review. Proposing clear steps for FDA/EMA approval, including essential validation benchmarks like analytical and clinical performance criteria, would offer valuable guidance on how to bring miRNA-based diagnostics to market. This would further emphasize the practical and translational potential of miRNA technologies in clinical applications.
Response: As the Food and Drug Administration (FDA) and European Medicines Agency (EMA) are adopting new treatment modalities, these institutions require accurate biomarkers to monitor disease progression and treatment efficacy. While clinical biomarkers are well-established and helpful in characterizing disease progression, there is a critical need for more robust and sensitive circulating biomarkers, such as nucleic acids and other small molecules. Used alone or in combination with clinical biomarkers, they will play a critical role in improving patient stratification for clinical trials and access to approved treatments, as well as in tracking response to therapy, paving the way for the development of individualized therapeutic approaches. A biomarker is defined as an objectively measured parameter that serves as an indicator of normal biological processes, pathological processes, or biological responses to a therapeutic intervention (as defined by the FDA or EMA). Biomarkers can be divided into susceptibility/risk assessors, diagnostic, monitoring, prognostic, predictive, response and safety biomarkers, and pharmacodynamic biomarkers.
Reviewer 2 Report
Comments and Suggestions for Authors
Comments and Suggestions for Authors are included in the attached file.

Author Response
Reviewer #2
Comments 1: nThe introduction discusses the heterogeneity of breast tumors and the limitations of current biomarkers. Could the authors elaborate on how miRNAs specifically address these limitations compared to traditional markers such as hormone receptors or HER2 status?
Response: The expression of microRNAs is more significant in patients with positive hormone receptors and lower in the basal-like subtype. Before and after chemotherapy, microRNAs expression is strongly associated with the expression of BRCA1 and BRCA2 genes, as well as with the expression of the p53 gene. This is not observed for HER2. Furthermore, the detection limits of HER2 are much lower than for microRNAs.
Comments 2:The authors mention the inclusion of eighty systematic reviews in their analysis. What criteria were used to ensure the quality and relevance of these reviews, and how was potential overlap between studies managed?
Response: Two investigators (WJ and SO) independently extracted data from eligible articles that met the inclusion and exclusion criteria. Discrepancies were then reviewed and resolved through discussion. Studies were included in the meta-analysis if they met the inclusion criteria.
Comments 3:The search strategy combined multiple databases and achieved a high recovery rate. Could the authors provide more details on the search period, specific inclusion/exclusion criteria, and whether a PRISMA flow diagram was used to document the selection process?
Response: The search period covered the years 2000 - 2025. The inclusion/exclusion criteria are given in the text and in the flow diagram (fig 1)
Comments 4: For studies included in the meta-analysis, were any methods used to assess publication bias or heterogeneity among the studies, and if so, what were the findings?
Response: Publication bias was not assessed in the meta-analysis. The objectivity of the publication was assumed, and the number of 1930 references, in our opinion, ensures the impartiality and heterogeneity of the studies.
Comments 5: The review states that miRNA expression profiles differ significantly between normal, precancerous, and cancerous breast tissues. Which specific miRNAs showed the most consistent changes across studies, and how robust are these findings given potential methodological differences among studies?
Response: The expressions of microRNA-21, microRNA-155, and microRNA -365 show the greatest changes between normal, precancerous, and cancerous tissues. In the case of these microRNAs, significant differences are also observed between the individual stages of cancer. Furthermore, after surgical resection, the levels of microRNA-21 and microRNA-155 were significantly decreased.
Comments 6: The manuscript mentions that miRNAs can be detected in various biological fluids. What is the comparative diagnostic performance (e.g., sensitivity, specificity) of tissue-based versus circulating miRNA assays for early breast cancer detection?
Response: The sensitivity and specificity of microRNA-155 in blood are 83.3% and 82.4%, respectively (AUC 0.856). The sensitivity and specificity of microRNA-121 in blood are 72.2% and 64.7%, respectively (AUC 0.699). The sensitivity and specificity of microRNA-583 in tumor tissue are 5.49% and 96.3%, respectively (AUC 0.69). The sensitivity and specificity of microRNA-877-5p in tumor tissue are 5.48% and 96.1%, respectively (AUC 0.63).
Comments 7: The authors describe the dual role of certain miRNAs as oncogenes or tumor suppressors. Are there examples where the same miRNA exhibits opposite functions in different breast cancer subtypes or stages, and how might this affect their utility as biomarkers?
Response: Some cancer-related miRNAs are also known as context-dependent miRNAs because they can exhibit specific effects dependent on tumor tissue, i.e., a single miRNA can have oncogenic or tumor-suppressive effects in different tumors. An example of such a dual-functional microRNA is miR-29, which suppresses lung tumors, cellular lymphomas, and oncogenic breast cancer. There is no evidence that the same microRNA is an oncogene or tumor suppressor in different breast cancer subtypes.
I have described this point in the text (pages 5-6):
Cancer-related miRNAs generally fall into two categories.The first group of oncogenic microRNAs (oncomiRs) includes highly expressed miRNAs.They are responsible for tumor progression and are important in maintaining the tumor phenotype.Tumor suppressive miRNAs (miRsupps) belong to the second group and inhibit carcinogenesis by regulating cell growth, apoptosis, immune cell proliferation, and other factors that affect tumorigenesis and that can often be downregulated in different tumors [26, 27].Some cancer-related miRNAs are also known as context-dependent miRNAs because they can exhibit specific effects dependent on the tumor tissue, i.e., a single miRNA can have oncogenic or tumor-suppressive effects in different tumors. An example of such a dual-function microRNA is miR-29, which suppresses lung tumors, cellular lymphomas, and oncogenic in breast cancer [28, 29]. By altering the expression of tumor suppressor genes and oncogenes, microRNAs can inhibit or exacerbate cancer progression depending on whether their expression is increased or decreased.This dual function highlights their potential as therapeutic targets as well as diagnostic tools. In this regard, specific microRNAs whose expression is reduced in atypical ductal hypoplasia but increased in invasive ductal carcinoma could be considered as early warning signals for cancer progression. Wang et al. were the first to perform a large-scale bioinformatic analysis of human oncogenic and suppressive microRNAs examining different functions, gene expression, evolutionary rate, molecular size, free energy, and transcription factors. They showed that oncogenic microRNAs more frequently than suppressive microRNAs cleave target mRNAs in tumor tissue. Moreover, oncomiRs coding sequences were observed primarily in amplified chromosomal regions, in contrast to miRsupp sequences, which were more frequently present in deleted chromosomal regions [30, 31]. However, although it has been confirmed that oncogenic microRNAs in cancer patients are up-regulated and suppressive microRNAs are down-regulated or disappear, the expression of some traditional oncomicroRNAs in patient blood decreased during the course of breast cancer. These results indicate that it is not possible to precisely define oncogenic or suppressive properties of individual microRNAs [32]. MiRNA expression profiles in various human cancers showed a common reduction in the pool of microRNAs relative to microRNAs present in normal tissues, Many cancer types had distinct miRNA expression profiles that could be used to distinguish tumor type or tissue origin of poorly differentiated tumors. The finding of multiple isoforms of microRNAs associated with cancer confirms that these isomiRNAs play an essential role in miRNA-mRNA regulatory networks and that changes in the expression profiles of isomicroRNAs lead to cancer development.
Comments 8: The review references the use of miRNA signatures for prognosis and therapy response. Could the authors summarize which miRNA panels have been most validated for these purposes in clinical studies?
Response: MicroRNAs have the potential to predict the response of breast cancer to systemic treatments. This includes the expression of microRNA-342-3p and microRNA-187-3p which was linked to systemic treatment success. High expression levels of miRNA-342-3p and miRNA-187-3p support progression-free and overall survival [80].
Comments 9: Were any limitations identified in the reviewed studies regarding sample size, patient diversity, or technical variability that could impact the generalizability of the results?
Response: Of course, the studies differed in the size of the studied groups, ethnic origin, and research methods, which on the one hand allows for generalization of the results, and on the other hand allows for their reference to a specific group.
Comments 10: The discussion highlights the potential of miRNAs as therapeutic targets. What are the current challenges in translating miRNA-based diagnostics or therapeutics into clinical practice, and how might these be addressed?
Response: It is common knowledge that chemotherapy and hormone therapy work through different mechanisms. Therefore, systemic treatment of breast cancer patients should be determined according to the molecular subtype. However, in the era of personalized therapy, more than one third of patients with breast cancer do not respond to systemic therapy. Therefore, ongoing efforts should be made to identify biomarkers that will help patients who will benefit from systemic therapy. Detection of a single microRNA or a group of microRNAs can predict resistance to multiple therapeutic strategies.
Comments 11: The authors note the high stability of miRNAs in biological fluids. Are there any pre-analytical or analytical factors (e.g., sample handling, normalization methods) that could affect the reproducibility of miRNA measurements in clinical settings?
Response: Undoubtedly, standardization is required to achieve repeatability of measurements, but there is a lack of data on specific pre-analytical or analytical factors that may affect measurements in clinical settings.
Comments 12: Given the complex regulatory networks involving miRNAs, how do the authors envision integrating miRNA biomarkers with other molecular or imaging modalities to improve breast cancer diagnosis and management?
Response: The demonstrated potential of microRNAs as biomarkers for breast cancer opens the path for future clinical trials. If this could be further developed in the lab-on-a-chip, it could prove to be a boon for women in low-income countries where women are reluctant to undergo physical examinations. A better understanding of microRNA targets through in silico analysis, extensive validation, and massive clinical trials opens up the possibility of more refined, cost-effective, and non-invasive methods in breast cancer diagnosis with a global reach of the world population. The next steps include rigorous validation of these microRNAs in larger groups of women, followed by the design and implementation of clinical trials to test their efficacy in clinical practice.
Round 2
Reviewer 1 Report
Comments and Suggestions for Authors
The authors have thoroughly addressed the comments raised by the previous reviewers, demonstrating careful attention to detail and a commitment to improving the manuscript. In my opinion, the article, in its current form, meets the standards of quality and scientific rigor expected for publication in the International Journal of Molecular Sciences journal